# Vortex loop operators and quantum M2-branes

**Nadav Drukker⋆ and Omar Shahpo†**

Department of Mathematics, King's College London, London, WC2R 2LS, United Kingdom

⋆ nadav.drukker@gmail.com , † omar.shahpo@kcl.ac.uk

## Abstract

We study M2-branes in $AdS_4 \times S^7/\mathbb{Z}_k$ dual to 1/2 and 1/3 BPS vortex loop operators in ABJM theory and compute their one-loop correction beyond the classical M2-brane action. The correction depends only on the parity of $k$ and is independent of all continuous parameters in the definition of the vortex loops. The result for odd $k$ agrees with the answers for the 1/2 BPS Wilson loop in the $k = 1$ theory and for even $k$ with the one in the $k = 2$ theory. Combining with the classical part, we find that the natural expansion parameter seems to be $1/\sqrt{kN}$ rather than $1/\sqrt{N}$. This provides a further setting where semiclassical quantisation can be applied to M2-branes and produces new results inaccessible by other methods.

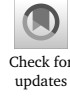

# 1 Introduction and conclusions

The holographic duality between M2-branes and observables in conformal field theories [1, 2] enables a deeper understanding of both. Field theory techniques such as localisation [3, 4] provide exact predictions for M2-brane partition functions. Those can be matched with classical M2-brane computations [5–12] and more recently semiclassical calculations [13–16].

Conversely, in cases when the field theory techniques are not as well developed, the M2-brane description may fill the gap by providing strong coupling results. For surface operators in the 6d (2,0) theory [17], M2-branes are crucial in computing their expectation values to leading [18–21] and subleading [22,23] orders at large $N$. In ABJM theory, M2-branes provide the holographic description of a class of 1d observables known as vortex loops [24]. In this paper, we extend the computation of their expectation values to one-loop using semiclassical M2-brane tools.

Vortex loops are operators supported on a closed contour on which some of the matter and gauge fields are singular. They are disorder operators, similar to 't Hooft or surface operators in 4d gauge theories. In pure Chern-Simons theory, they are equivalent to Wilson loops [25], while they have richer features in Chern-Simons-matter and other interacting 3d theories. They were described in ABJM theory in [24] and the mapping between them and Wilson loops under 3d $\mathcal{N} = 4$ mirror symmetry was explained in [26]. Particular examples of abelian vortices in $\mathcal{N} = 2$ were computed using supersymmetric localisation in [27,28].

We study the holographic description of 1/2 and 1/3 BPS vortex loops in ABJM theory. The simplest class are characterised on the field theory side by the gauge symmetry breaking to $U(N-1)^2 \times U(1)$ [24]. In coordinates $(t, z, \bar{z})$ on $\mathbb{R}^3$ with $z$ a complex coordinate on the spatial slice, placing the vortex line at $z = 0$, the four ABJM bifundamental matter fields $C^I$ take the singular classical configuration

$$
C^1 = \frac{1}{\sqrt{z}} \begin{pmatrix} 0_{N-1} & 0 \\ 0 & \beta_1 \end{pmatrix}, \qquad C^2 = \frac{1}{\sqrt{\bar{z}}} \begin{pmatrix} 0_{N-1} & 0 \\ 0 & \beta_2 \end{pmatrix}, \qquad C^3 = C^4 = 0, \tag{1}
$$

where $\beta_{1,2}$ are complex parameters, while the two gauge fields of ABJM $A$ and $\hat{A}$ are given by

$$
A_z = \hat{A}_z = -\frac{i}{4kz} \begin{pmatrix} 0_{N-1} & 0 \\ 0 & \alpha \end{pmatrix}, \qquad A_t = \hat{A}_t = -2\pi(C^1 C_1^\dagger - C^2 C_2^\dagger), \tag{2}
$$

where $\alpha$ is an angular variable. This configuration is 1/3 BPS, with supersymmetry enhanced to 1/2 when $\beta_2 = 0$.

At the quantum level, a vortex loop is defined by the path integral with this singular behaviour near $z = 0$. The definition above is for the infinite straight line, but to get a finite expectation value we should really look at the circle, which is just a conformal transformation of the above configuration, as is familiar in the case of Wilson loops [29].

The vortex loop (1) is described in $AdS_4 \times S^7/\mathbb{Z}_k$ by a single M2-brane with $AdS_2 \times S^1$ geometry [24]. For the background we take the metric

$$
ds^2 = \frac{R^2}{4} \left( du^2 + \cosh^2 u\, h_{AdS_2} + \sinh^2 u\, d\phi^2 \right) + R^2 ds^2_{S^7/\mathbb{Z}_k}, \tag{3}
$$

and $S^7/\mathbb{Z}_k$ can be written as a $U(1)$ fibre over $\mathbb{CP}^3$, see (6). This choice of metric is an $AdS_2 \times S^1$ foliation of $AdS_4$ and its boundary at $u \to \infty$ has the geometry $AdS_2 \times S^1$. This metric is conformal to $\mathbb{R}^3$ such that under the conformal transformation, the boundary of $AdS_2$ is mapped to a line or a circle. Therefore, this metric is ideal for studying conformal circle and line operators. More specifically, the boundary of $AdS_2$ for all values of $u$ are mapped to the same line/circle.

The M2-branes are extended along $AdS_2$ at fixed $u = u_0$. The third direction wraps the $\phi$ circle while also following the fibre direction (and in the case of the 1/3 BPS loop, also a circle on the $\mathbb{CP}^3$ base). All cycles are mutually periodic when the fibre direction is wrapped $k$ times and the others are wrapped twice. For even $k$ it suffices then to have $k/2$ periods, which was shown in [24] to match the properties of the field theory vortex at even $k$.

This holographic description was used in [24] to compute the vortex loop expectation value and some correlation functions at strong coupling. As reviewed in Section 2.1, the answer is given by the classical M2-brane action

$$\log\langle V\rangle = -S^{(0)} = \begin{cases} 2\pi\sqrt{kN/2} + O(N^0) & k \text{ odd.} \\ \pi\sqrt{kN/2} + O(N^0) & k \text{ even.} \end{cases} \tag{4}$$

The result is independent of $u_0$ (or $\beta_1$ in field theory language, c.f. (12)) and is also the same for any of the 1/3 BPS loops (so arbitrary $\beta_2$). In the limit of $u_0 = 0$ the brane configuration agrees with that describing $k/2$ or $k$ coincident Wilson loops. Unsurprisingly, the classical answer is $k/2$ or $k$ times the answer for the 1/2 BPS Wilson loop [30].

More precisely, the $k$ coincident M2-branes should correspond to a Wilson loops in some (possibly reducible) $k$-dimensional representation of $U(N)$ (or really $U(N|N)$ for the 1/2 BPS loop [30]). Recall that monopole operators in Chern-Simons theories transform in the $k$-dimensional symmetric representation [2,31,32]. This has the effect that charge can change by $k$ units and is sometimes referred to as the bosonic exclusion principle [33]. This may be behind the fact that the $k$-dimensional Wilson loop may be continuously deformed into a vortex loop with arbitrary $u_0$.

To compute the one-loop correction to the classical action (4), we study the M2-brane's quadratic fluctuation action. This follows on the study of fluctuations of classical string solutions in $AdS_5 \times S^5$ [34], and of M2-branes in [22,35,36]. This is the main computation of this paper and is in Sections 2 and 3 below.

We find that the quadratic action is identical in both the 1/2 and 1/3 BPS cases and depends on the parameter $u_0$ only via overall prefactors. One can absorb these factors by rescaling the fluctuating fields and in any case, it does not affect the computation of the one-loop determinant. The action (21), (35) contains explicit dependence on $k$, but this can be removed by rescaling the coordinate $\zeta$ and bringing its range back to $[0, 2\pi]$. So the answer is really also insensitive to the value of $k$, only to whether it is even or odd.

We therefore find that as with the classical calculation, the quadratic action agrees with that for the 1/2 BPS Wilson loop in the $k = 1$ and $k = 2$ theories, as derived in [36]. The determinants of these actions were evaluated in [13] by Kaluza-Klein reduction on the $S^1$ circle and using known results for determinants on $AdS_2$ [34,37–39]. The results for the single Wilson loop depends on the value of $k$, but we only need to borrow the results for $k = 1, 2$ which are respectively $-\log 4$ and $0$. So we find that (4) gets corrected at one-loop to

$$\log\langle V\rangle = \begin{cases} 2\pi\sqrt{kN/2} - \log 4 + O(N^{-1/2}), & k \text{ odd,} \\ \pi\sqrt{kN/2} + 0 + O(N^{-1/2}), & k \text{ even.} \end{cases} \tag{5}$$

In the absence of an exact computation of the expectation value of the vortex loop we may wish to learn more general lessons from the classical and semiclassical answer.

First, we see that there is no dependence on $u_0$ and it is natural to conjecture that this persists to all orders in the $1/\sqrt{N}$ expansion. $u_0$ does not appear at all in the classical action and can be absorbed into the definition of the quadratic fluctuation fields. If a similar mechanism extends to the interacting theory, there will be no $u_0$ dependence.

Second, the dependence on $k$, apart for the separation into odd and even cases appears together with $N$, so the natural expansion parameter seems to be $1/\sqrt{kN}$, rather than $1/\sqrt{N}$.

This combination is related to the radius of space (see (9)) and is insensitive to the order of the orbifold $k$. This is not surprising, as the solution wraps the orbifold $k$ or $k/2$ times, so is natural to study it in the covering space. Indeed, for this perturbative calculation, it is not clear how the order of the orbifold could show up, as for $u_0 \neq 0$ the M2-brane never crosses itself and the local geometry is always like in the $k = 1, 2$ case.

If we focus on the theories with $k = 1, 2$, those have enhanced $\mathcal{N} = 8$ supersymmetry and the vortex loop preserves 16 supercharges. In fact, the 1/3 BPS loop is now enhanced to 1/2 BPS, as $SO(8)$ symmetry allows to rotate $C_2^\dagger$ into $C^1$. Then, if indeed the $k$ dependence factors with $N$, this result extends to all $k$.

It is natural to conjecture that vortex loops are described exactly by the 1/2 BPS Wilson loop in the $k = 1, 2$ theory, or that they differ by non-perturbative corrections. This requires either exact field theory calculations or the study of the interaction of the M2-branes describing the vortex with M2-brane instantons.

Unfortunately, $k = 1, 2$ are the only cases when the exact expectation value of the 1/2 BPS circular Wilson loop is not known. The expectation value of the Wilson loop was evaluated from the localisation matrix model at finite $k$ and large $N$ in [40]. This was extended to all orders in $N$ in [41] (see also [42–44]). Unfortunately, the resulting expression diverges for $k = 1, 2$. Therefore, there is no way to try to borrow the Wilson loop result and try to make a conjecture for the vortex loop. A more careful localisation calculation is required for $k = 1, 2$ and this would hopefully supply the exact answer for both the Wilson and vortex loop.

## 2 The 1/2 BPS configuration

The vortex operators (1) are described by single M2-branes in $AdS_4 \times S^7/\mathbb{Z}_k$, with the metric (3) in the introduction.[1] We write the metric on $S^7/\mathbb{Z}_k$ as a $U(1)$ bundle over $\mathbb{CP}^3$

$$
\begin{aligned}
ds^2_{S^7/\mathbb{Z}_k} &= ds^2_{\mathbb{CP}^3} + \frac{1}{k^2}(kA + d\zeta)^2 \,, \\
ds^2_{\mathbb{CP}^3} &= \frac{(1 + w^m \bar{w}^{\bar{m}}) dw^n d\bar{w}^{\bar{n}} - \bar{w}^{\bar{n}} w^m dw^n d\bar{w}^{\bar{m}}}{(1 + w^k \bar{w}^{\bar{k}})^2} \,.
\end{aligned}
\tag{6}
$$

where $w^n$ are complex coordinates, with the bar denoting complex conjugation, and $\zeta$ has $2\pi$ period. $A$ is a real one form

$$
A = -\frac{i}{2} \frac{\bar{w}^{\bar{n}} dw^n - w^n d\bar{w}^{\bar{n}}}{1 + w^m \bar{w}^{\bar{m}}} \,.
\tag{7}
$$

The Kähler form on $\mathbb{CP}^3$ is then $K = dA/2$.

The background also has a four-form $F_4$ proportional to the volume form on $AdS_4$, with a potential $C_3$, such that

$$
C_3 = \frac{R^3}{8}(\cosh^3 u - 1)\Omega_{AdS_2} \wedge d\phi \,, \qquad F_4 = dC_3 = \frac{3R^3}{8}\Omega_{AdS_4} \,.
\tag{8}
$$

The choice of gauge for $C_3$ is compatible with the $AdS_2$ symmetry of the problem. This is actually crucial in the calculation of both the classical action and the fluctuations due to the fact that the brane is non-compact.

The relation between the gravity quantities and those of the field theory are

$$
\frac{R^3}{l_{\text{pl}}^3} = 4\pi\sqrt{2kN} \,,
\tag{9}
$$

where $l_{\text{pl}}$ is Planck length.

---

[1] For the indices, we use $\mu, \nu$ to indicate $AdS_4$ indices, $a, b$ for $AdS_2 \subset AdS_4$, $m, n$ for $\mathbb{CP}^3$ indices, and 7 for the $U(1)$ fibre. $A, B$ are used for the full 11d space and $i, j$ for the 3d world-volume. We employ hats to indicate flat tangent space indices.

## 2.1 Classical M2-brane solution

The classical M2-brane action [45] has a Nambu-Goto part and a Wess-Zumino part coupling the embedding of the M2-brane to the pullback (indicated by a star) of the 3-form $C_3$

$$S = T_{\text{M2}} \int (\Omega_{\text{M2}} + {}^*C_3),$$ 

(10)

where $\Omega_{\text{M2}}$ is the volume form on the M2-brane. $T_{\text{M2}} = 1/4\pi^2 l_{\text{pl}}^3$ is the M2-brane tension.

The M2-brane dual to the 1/2 BPS vortex (1) with $\beta_2 = 0$ is extended in $AdS_2$ at fixed $u_0$. The remaining direction is a circle embedded in both $AdS_4$ and $S^7/\mathbb{Z}_k$. Parameterising it in terms of $\zeta$ from (6), the solution is [24]

$$u = u_0, \qquad \phi = \frac{2}{k}\zeta + \phi_0, \qquad w^m = w_0^m,$$ 

(11)

where $u_0$, $\phi_0$ and $w_0^m$ are constants. Clearly to close the $\phi$ circle (for $u_0 \neq 0$), we need to wrap the $\zeta$ circle $k$ times (or $k/2$ for even $k$).

$w_0^m$ correspond to the choice of scalars that are turned on in (1), and $u_0$ and $\phi_0$ are fixed by

$$\sinh u_0 = \frac{1}{\pi}\sqrt{\frac{k}{2N}}|\beta_1|, \qquad \phi_0 = -\frac{4\pi\alpha}{k}.$$ 

(12)

For simplicity, we set $w_0^m = \phi_0 = 0$.

We note that in the limit $\beta_1 \to 0$, we recover the M2-brane configuration dual to the 1/2 BPS Wilson loop [13, 30], though wrapped $k$ or $k/2$ times over the $\zeta$ circle, rather than once as for a fundamental Wilson loop.

The induced metric on the M2-brane is

$$ds^2 = \frac{R^2 \cosh^2 u_0}{4}h_0, \qquad h_0 = h_{AdS_2} + \frac{4}{k^2}d\zeta^2.$$ 

(13)

Including the pullback of $C_3$ (8), the classical action $S^{(0)}$ is

$$S^{(0)} = \frac{T_{M2}R^3}{4k}\int \Omega_{AdS_2}d\zeta = \begin{cases} -T_{M2}R^3\pi^2, & k \text{ odd}, \\ -T_{M2}R^3\pi^2/2, & k \text{ even}, \end{cases}$$ 

(14)

where $\Omega_{AdS_2}$ is the $AdS_2$ volume form, and we have used that the regularised volume for $AdS_2$ is $-2\pi$. This gives the classical expectation values (4). This is $k$ (for odd $k$) or $k/2$ (for even $k$) times the expectation value of the 1/2 BPS Wilson loop [30, 42].

## 2.2 Bosonic fluctuations

To compute the one-loop corrections, we need to integrate over quadratic fluctuations to the M2-brane action. At quadratic order, the bosonic and fermionic fluctuations decouple.

The bosonic part of the fluctuation action is obtained by expanding the action (10) around the classical solution (11). We take as world-volume coordinates $\sigma^a$ for $AdS_2$ and $\zeta$ and expand the other coordinates around their classical values

$$u(\sigma^a, \zeta) = u_0 + \eta(\sigma^a, \zeta), \qquad \phi(\sigma^a, \zeta) = \frac{2}{k}\zeta + \varphi(\sigma^a, \zeta), \qquad w^m(\sigma^a, \zeta).$$ 

(15)

$\eta$, $\varphi$, $w^m$ and $\bar{w}^{\bar{m}}$ are the fluctuation fields. The one-form $A$ in (7) appearing in the background metric is to quadratic order

$$A \simeq \frac{i}{2}(w^m d\bar{w}^{\bar{m}} - \bar{w}^{\bar{m}}dw^m).$$ 

(16)

With this, the metric induced from (3) is to quadratic order

$$
\begin{aligned}
ds^2 \simeq \frac{R^2}{4}\bigg( &\big((\cosh^2 u_0(1+2\eta^2) + 2\eta\sinh u_0\cosh u_0 - \eta^2\big)h_0 \\
&+ \big(\partial_i\eta\partial_j\eta + \sinh^2 u_0\,\partial_i\varphi\partial_j\varphi + 4\partial_i w^m\partial_j\bar{w}^{\bar{m}}\big)d\sigma^i d\sigma^j \\
&+ \frac{4}{k}\big(\sinh^2 u_0\,\partial_i\varphi + (iw^m\partial_i\bar{w}^{\bar{m}} - i\bar{w}^{\bar{m}}\partial_i w^m)\big)d\sigma^i d\zeta\bigg).
\end{aligned}
\tag{17}
$$

This gives the quadratic correction to the Nambu-Goto action

$$
\begin{aligned}
S_{\text{NG}}^{(2)} = \frac{R^3}{16}T_{\text{M2}}\cosh u_0\int\sqrt{h_0}\bigg(&h_0^{ij}(\partial_i\eta\partial_j\eta + \tanh^2 u_0\,\partial_i\varphi\partial_j\varphi + 4\partial_i w^m\partial_j\bar{w}^{\bar{m}}) \\
&+ 3(3\cosh^2 u_0 - 2)\eta^2 + k(3\cosh^2 u_0 - 1)\tanh u_0\,\eta\partial_\zeta\varphi + \frac{ik}{4}(w^m\partial_\zeta\bar{w}^{\bar{m}} - \bar{w}^{\bar{m}}\partial_\zeta w^m)\bigg).
\end{aligned}
\tag{18}
$$

The pullback of the three form $C_3$ to quadratic order is

$$
{}^*C_3^{(2)} = -\frac{R^3\sqrt{h_0}\cosh u_0}{16}\big(3\cosh u_0\sinh u_0\,\eta\partial_\zeta\varphi + 3(3\cosh^2 u_0 - 2)\eta^2\big).
\tag{19}
$$

We define a further complex field $\chi$ as

$$
\chi = \frac{1}{2}(\eta - i\tanh u_0\,\varphi).
\tag{20}
$$

Combining (18) and (19), we obtain the quadratic bosonic action

$$
\begin{aligned}
S^{(2)} = \frac{R^3}{4}T_{\text{M2}}\cosh u_0\int d^3\sigma\sqrt{h_0}\bigg(&h_0^{ij}(\partial_i\bar{\chi}\partial_j\chi + \partial_i w^m\partial_j\bar{w}^{\bar{m}}) \\
&+ \frac{ik}{4}(\chi\partial_\zeta\bar{\chi} - \bar{\chi}\partial_\zeta\chi + w^m\partial_\zeta\bar{w}^{\bar{m}} - \bar{w}^{\bar{m}}\partial_\zeta w^m)\bigg).
\end{aligned}
\tag{21}
$$

$k$ dependence appears explicitly in the second line of (21), but also in the range of the $\zeta$ coordinate (see the comment under (11)). Rescaling $\zeta$ to the range $[0, 2\pi]$ replaces the $k$ in the second line by 1 or 2. For $u_0 = 0$ the resulting action is identical to that derived in [36] for M2-brane describing the 1/2 BPS Wilson loop in the $k = 1$ and $k = 2$ theories. The overall $\cosh u_0$ factor can be absorbed in a rescaling of the fluctuating fields and does not affect the determinant arising from the path integral. The result then agrees with the calculation of the determinant in [13], which combined with the fermionic contribution gives the answer in (5).

## 2.3 Fermionic fluctuations

To write the action for the fermionic coordinates on the M2-brane, we need a vielbein for the target space.[2] A choice of vielbein (written as forms) is

$$
Re^{\hat{a}} = \frac{R\cosh u}{2}\hat{e}^{\hat{a}}, \quad Re^{\hat{2}} = \frac{R\sinh u}{2}d\phi, \quad Re^{\hat{3}} = \frac{R}{2}du, \quad Re^{\hat{7}} = \frac{R}{k}(d\zeta + kA), \quad Re^{\hat{m}}, \quad Re^{\bar{\hat{m}}},
\tag{22}
$$

where $\hat{e}^{\hat{a}}$ are a vielbein of the unit radius $AdS_2$, and $e^{\hat{m}}, e^{\bar{\hat{m}}}$ are three pairs of complex conjugate vielbeine of $\mathbb{CP}^3$, written explicitly in (53), such that

$$
ds^2_{\mathbb{CP}^3} = \kappa_{\hat{m}\bar{\hat{n}}}e^{\hat{m}}e^{\bar{\hat{n}}} + \kappa_{\bar{\hat{m}}\hat{n}}e^{\bar{\hat{m}}}e^{\hat{n}},
\tag{23}
$$

---

[2]We perform the calculation in Lorentzian signature, and Wick-rotate in the end.

with $\kappa$ the Kähler metric on $\mathbb{C}^3$, explicitly $\kappa_{\hat{m}\bar{\hat{n}}} = \kappa_{\bar{\hat{m}}\hat{n}} = \delta_{mn}/2$, which is used to lower and raise the indices $\hat{m}$ and $\bar{\hat{m}}$.

The spin connection for this frame is

$$\Omega^{\hat{2}}{}_{\hat{3}} = \cosh u \, d\phi\,, \qquad \Omega^{\hat{a}}{}_{\hat{3}} = \sinh u \, \hat{e}^{\hat{a}}\,, \qquad \Omega^{\hat{0}}{}_{\hat{1}}\,,$$
$$\Omega^{\hat{m}}{}_{\hat{n}} = -\Omega^{\bar{\hat{n}}}{}_{\bar{\hat{m}}} = \hat{\Omega}^{\hat{m}}{}_{\hat{n}} + i\delta^{\hat{m}}_{\hat{n}} e^{\hat{7}}\,, \qquad \Omega^{\hat{7}}{}_{\hat{m}} = K_{\hat{m}\bar{\hat{n}}} e^{\bar{\hat{n}}}\,, \qquad \Omega^{\hat{7}}{}_{\bar{\hat{m}}} = -K_{\hat{n}\bar{\hat{m}}} e^{\hat{n}}\,, \tag{24}$$

where $\hat{\Omega}^{\hat{m}}{}_{\hat{n}}$ is the spin connection of unit radius $\mathbb{CP}^3$, (56) [46]. $\Omega^{\hat{0}}{}_{\hat{1}}$ is the $AdS_2$ spin connection of the frame $\hat{e}^{\hat{a}}$, and $K_{\hat{m}\hat{n}}$ are the components of the Kähler two-form in this frame, i.e. $K = dA/2 = K_{\hat{m}\bar{\hat{n}}} e^{\hat{m}} \wedge e^{\bar{\hat{n}}}$, and are

$$K_{\hat{m}\bar{\hat{n}}} = \frac{i}{2}\delta_{\hat{m}\hat{n}}\,. \tag{25}$$

The spin connection for $\mathbb{CP}^3$ with the $U(1)$ fibre is discussed in more detail in Appendix A.

The action for fermions at quadratic order involves a 32-component Majorana spinor $\theta$ and takes the form [45, 47, 48]

$$S_F^{(2)} = -\frac{T_{M2}R^2\cosh^2 u_0}{4} \int d^3\sigma \sqrt{-h_0}\, \bar{\theta}\gamma^i(1-\gamma)D_i\theta\,, \tag{26}$$

where $\bar{\theta} \equiv \theta^\dagger\Gamma_{\hat{0}} = \theta^\mathsf{T} C$, with $C$ being the charge conjugation matrix, $h_0$ is given in (13), and $\gamma^i$ are world-volume gamma matrices expressed in terms of the 11d matrices (after the rescaling in (13)) as

$$\gamma^i = \frac{\cosh u_0}{2}\,{}^*e^i_{\hat{A}}\Gamma^{\hat{A}}\,, \qquad \gamma = \frac{1}{3!\sqrt{-h_0}}\epsilon^{ijk}\gamma_{ijk}\,. \tag{27}$$

Here ${}^*e^i_{\hat{A}}$ is the pullback of the vielbein and $\epsilon^{ijk}$ is the Levi-Civita symbol with $\epsilon^{123} = 1$.

Using the classical solution (11) and our choice of vielbeine (22), the world-volume gamma matrices explicitly are

$$\gamma^a = \hat{e}^a_{\hat{b}}\,\Gamma^{\hat{b}}\,, \quad \gamma^\zeta = \frac{k}{2\cosh u_0}\left(\Gamma^{\hat{2}}\sinh u_0 + \Gamma^{\hat{7}}\right)\,, \quad \gamma = \frac{1}{\cosh u_0}\Gamma_{\hat{0}}\Gamma_{\hat{1}}\left(\Gamma_{\hat{2}}\sinh u_0 + \Gamma_{\hat{7}}\right)\,, \tag{28}$$

The 11d gamma matrices are those for the frame (22) and satisfy the complexified Clifford algebra

$$\{\Gamma_{\hat{\mu}},\Gamma_{\hat{\nu}}\} = 2\eta_{\mu\nu}\,, \qquad \{\Gamma_{\hat{m}},\Gamma_{\bar{\hat{n}}}\} = 2\kappa_{\hat{m}\bar{\hat{n}}}\,, \qquad (\Gamma_{\hat{7}})^2 = 1\,, \tag{29}$$

with all other anti-commutators vanishing. The world-volume ones then satisfy the usual anti-commutation relations

$$\{\gamma^i,\gamma^j\} = 2h_0^{ij}\,. \tag{30}$$

The covariant derivatives $D_i$ are defined as

$$D_i = \partial_i + \frac{1}{4}{}^*\Omega_i^{\hat{A}\hat{B}}\Gamma_{\hat{A}\hat{B}} - \frac{1}{2}\left({}^*e_i^{\hat{\mu}}\Gamma^{\hat{0}\hat{1}\hat{2}\hat{3}}\Gamma_{\hat{\mu}} + {}^*e_i^{\hat{A}}\Gamma^{\hat{0}\hat{1}\hat{2}\hat{3}}\Gamma_{\hat{A}}\right)\,, \tag{31}$$

noting that the second spin connection index is raised and we define $\Gamma_{\hat{m}\bar{\hat{n}}} = \frac{1}{2}\left(\Gamma_{\hat{m}}\Gamma_{\bar{\hat{n}}} - \Gamma_{\bar{\hat{n}}}\Gamma_{\hat{m}}\right)$ and similarly $\Gamma_{\bar{\hat{n}}\hat{m}}$. For our configuration, these are explicitly

$$D_a = \partial_a - {}^*e_a^{\hat{a}}\Gamma^{\hat{0}\hat{1}\hat{2}\hat{3}}\Gamma_{\hat{a}} + \frac{1}{2}{}^*\Omega_a^{\hat{0}\hat{1}}\Gamma_{\hat{0}\hat{1}} + \frac{\sinh u_0}{2}{}^*e_a^{\hat{a}}\Gamma_{\hat{a}\hat{3}}\,,$$
$$D_\zeta = \partial_\zeta - \frac{\sinh u_0}{2k}\Gamma^{\hat{0}\hat{1}\hat{2}\hat{3}}\Gamma_{\hat{2}} - \frac{1}{2k}\Gamma^{\hat{0}\hat{1}\hat{2}\hat{3}}\left(\Gamma_{\hat{2}}\sinh u_0 + \Gamma_{\hat{7}}\right) + \frac{1}{2k}\left(2\cosh u_0\Gamma_{\hat{2}\hat{3}} + i\kappa^{\hat{m}\bar{\hat{n}}}\Gamma_{\hat{m}\bar{\hat{n}}}\right)\,. \tag{32}$$

Substituting in the expression for the gamma matrices (28) and the covariant derivatives (32) leads to a somewhat complicated action. However, as was done in the context of strings

in [34, 49], we can simplify it significantly by rotating the spinors such that the world-volume matrices $\gamma^i$ become constant. Such a rotation is equivalent to choosing an alternative vielbein to match the classical brane, as is done in Appendix A.2 of [23] and also implemented for the 1/3 BPS vortex loop solution in Section 3 below.

The world-volume matrix $\gamma^\zeta$ (28) is proportional to $\Gamma_{\hat{2}} \sinh u_0 + \Gamma_{\hat{7}}$. We thus define a rotation matrix $S$ such that

$$S^{-1}\Gamma_{\hat{7}}S = \frac{1}{\cosh u_0}\left(\Gamma_{\hat{2}}\sinh u_0 + \Gamma_{\hat{7}}\right), \qquad S^{-1}\Gamma_{\hat{2}}S = \frac{1}{\cosh u_0}\left(\Gamma_{\hat{2}} - \Gamma_{\hat{7}}\sinh u_0\right). \tag{33}$$

With this transformation, $S\gamma^\zeta S^{-1}$ is proportional to $\Gamma_{\hat{7}}$ and $\gamma$ is transformed as $S\gamma S^{-1} = \Gamma_{\hat{0}\hat{1}\hat{7}}$. Defining the transformed spinor $\vartheta \equiv S\theta$, and imposing the kappa fixing equation

$$\frac{1}{2}\left(1 + \Gamma_{\hat{0}\hat{1}\hat{7}}\right)\vartheta = 0\,, \tag{34}$$

we can replace $(1 - \gamma)\vartheta = 2\vartheta$ and find the fermionic action (26) explicitly is

$$S_F^{(2)} = -\frac{R^2 T_{M2}\cosh^2 u_0}{2}\int d^3\sigma\sqrt{-h_0}\,\bar{\vartheta}\left(\left(\gamma^a\partial_a + \frac{1}{2}{}^*\Omega_a^{\hat{0}\hat{1}}\gamma^a\Gamma_{\hat{0}\hat{1}}\right) + \frac{k}{2}\Gamma^{\hat{7}}\partial_\zeta \right.$$
$$\left. + \frac{5}{4}\Gamma^{\hat{0}\hat{1}\hat{2}\hat{3}} + \frac{5}{2}\Gamma_{\hat{3}}\sinh u_0 + \frac{i}{4}\kappa^{\hat{m}\bar{n}}\Gamma^{\hat{7}}\Gamma_{\hat{m}\bar{n}}\right)\vartheta\,, \tag{35}$$

where the first line has the form of a kinetic term of a 3d spinor on $AdS_2 \times S^1$ with metric $h_{0ab}$.

The term proportional to $\Gamma_{\hat{3}}$ drops out from the action because of the kappa fixing condition. Thus, the dependence of the action on $u_0$ is only through the overall prefactor $\cosh^2 u_0$, similar to the case of the bosonic action (21), though the prefactor there is $\cosh u_0$.

To further simplify the mass term on the second line we use the relation for our representation of the gamma matrices $\Gamma^{\hat{7}} = 8i\Gamma_{\hat{0}\hat{1}\hat{2}\hat{3}}\Gamma_{\hat{4}\bar{4}}\Gamma_{\hat{5}\bar{5}}\Gamma_{\hat{6}\bar{6}}$ to write it as

$$-\frac{i}{4}\Gamma^{\hat{7}}\left(40\Gamma_{\hat{4}\bar{4}}\Gamma_{\hat{5}\bar{5}}\Gamma_{\hat{6}\bar{6}} + 2\Gamma_{\hat{4}\bar{4}} + 2\Gamma_{\hat{5}\bar{5}} + 2\Gamma_{\hat{6}\bar{6}}\right). \tag{36}$$

We can easily diagonalise this matrix noting that $2\Gamma_{\hat{4}\bar{4}}$, $2\Gamma_{\hat{5}\bar{5}}$ and $2\Gamma_{\hat{6}\bar{6}}$ all square to 1, commute with each other and are self-adjoint so that their eigenspinors are orthonormal and complete. For a simultaneous eigenspinor $\vartheta_{\alpha_4\alpha_5\alpha_6}$ of the three operators with eigenvalues $\alpha_4$, $\alpha_5$ and $\alpha_6$ respectively, this matrix evaluates to

$$-\frac{i}{4}\Gamma^{\hat{7}}\left(40\Gamma_{\hat{4}\bar{4}}\Gamma_{\hat{5}\bar{5}}\Gamma_{\hat{6}\bar{6}} + 2\Gamma_{\hat{4}\bar{4}} + 2\Gamma_{\hat{5}\bar{5}} + 2\Gamma_{\hat{6}\bar{6}}\right)\vartheta_{\alpha_4\alpha_5\alpha_6} = -\frac{i}{4}\Gamma^{\hat{7}}\left(5\alpha_4\alpha_5\alpha_6 + \alpha_4 + \alpha_5 + \alpha_6\right)\vartheta_{\alpha_4\alpha_5\alpha_6}\,. \tag{37}$$

The eigenvalues are then $\pm 2i$ with degeneracy one and $\pm i$ with degeneracy 3.

This exactly matches the fermionic action found in [36] with $k = 1$ for odd $k$ and $k = 2$ for even $k$. The determinant of this differential operator was evaluated in [13] and together with the bosonic part give (5).

## 3 The 1/3 BPS configuration

The 1/2 BPS configuration (11) describes an M2-brane embedded at a point in $\mathbb{CP}^3$. A generalisation of these solutions is obtained by extending the M2-brane along a circle in $\mathbb{CP}^1 \subset \mathbb{CP}^3$, leading to a 1/3 BPS configuration [24]. These represent vortex loops (1) with non-zero $\beta_2$.

They are extended along an $AdS_2 \subset AdS_4$, whose coordinates we take as world-volume coordinates along with $\xi$, which is proportional to the phase of the $\mathbb{CP}^1$ coordinate $w^4$.[3] The M2-brane has the following classical embedding coordinates

$$\zeta = \xi + \zeta_0, \qquad u = u_0, \qquad \phi = \frac{2}{k}\xi + \phi_0, \qquad w^4 = re^{-2i\xi/k}, \qquad \bar{w}^{\bar{4}} = re^{2i\xi/k}, \qquad (38)$$

$\xi$ has the range $[0, \pi k]$ for even $k$, and $[0, 2\pi k]$ if $k$ is odd. Closing in either case the $\zeta$, $\phi$ and $w_4$ circles.

The constants in (38) are related to the vortex loop parameters in (1) by

$$\sinh u_0 = \frac{1}{\pi}\sqrt{\frac{k}{2N}}\sqrt{|\beta_1|^2 + |\beta_2|^2}, \qquad \phi_0 = 2\pi\alpha, \qquad re^{-i\zeta_0} = \frac{\beta_2}{\beta_1}. \qquad (39)$$

As before, using the symmetries of the M2-brane action, we can set $\phi_0 = \zeta_0 = 0$.

With this solution, the pullback metric is the same as in the 1/2 BPS case (13) and the classical action is the same as in (14) [24]. We turn next to computing the action for the quadratic bosonic fluctuations around this classical solution.

### 3.1 Bosonic fluctuations

In order to have a simple action for the perturbations, we take the fluctuations to be normal to the classical world-volume and orthonormal to each-other. We denote the fluctuating fields $y_2$, $y_3$, $y_m$, and $\bar{y}_{\bar{m}}$ such that the embedding is

$$u = u_0 + 2y_3,$$

$$\phi = \frac{2}{k}\xi - \frac{2}{\sinh u_0 \cosh u_0}y_2, \qquad \zeta = \xi + k\tanh u_0\, y_2 + \frac{irk}{2}(\bar{y}_{\bar{4}} - y_4),$$

$$w^4 = e^{-2i\xi/k}(r + y_4 + r^2\bar{y}_{\bar{4}} - 2iry_2), \qquad \bar{w}^{\bar{4}} = e^{2i\xi/k}(r + r^2 y_4 + \bar{y}_{\bar{4}} + 2iry_2), \qquad (40)$$

$$w^5 = \sqrt{1 + r^2}\, y_5, \qquad \qquad \bar{w}^{\bar{5}} = \sqrt{1 + r^2}\, \bar{y}_{\bar{5}},$$

$$w^6 = \sqrt{1 + r^2}\, y_6, \qquad \qquad \bar{w}^{\bar{6}} = \sqrt{1 + r^2}\, \bar{y}_{\bar{6}},$$

For $r = 0$ this matches the 1/2 BPS case (15) with the identifications $y_4 e^{-2i\xi/k} \to w^4$, $\bar{y}_4 e^{2i\xi/k} \to \bar{w}^{\bar{4}}$ and other obvious maps.

The pullback metric, written to second order in the perturbations is

$$ds^2 \simeq \frac{R^2}{4}\Bigg((\cosh^2 u_0(1 + 8y_3^2) + 4y_3 \sinh u_0 \cosh u_0 - 4y_3^2)h_0 \qquad (41)$$

$$+ 4\big(\partial_i y_3 \partial_j y_3 + 4\partial_i y_2 \partial_j y_2 + 4\partial_i y_m \partial_j \bar{y}_{\bar{m}}\big)d\sigma^i d\sigma^j + \frac{4i}{k}\big(\bar{y}_{\bar{m}}\partial_i y_m - y_m \partial_i \bar{y}_{\bar{m}} + 8iy_3 \partial_i y_2\big)d\xi d\sigma^i$$

$$+ \frac{4r}{k}\big(ir\big(y_4 \partial_i \bar{y}_{\bar{4}} - \bar{y}_{\bar{4}}\partial_i y_4\big) - 2\tanh u_0\big((y_4 + \bar{y}_{\bar{4}})\partial_i y_2 + (\partial_i y_4 + \partial_i \bar{y}_{\bar{4}})y_2\big)\big)d\xi d\sigma^i\Bigg),$$

where $h_0$ is defined in (13). We see that classically, the metric in the 1/2 BPS (17) and 1/3 BPS cases are the same. The quadratic term in the Nambu-Goto action is

$$S_{NG}^{(2)} = \frac{R^3}{4}T_{M2}\cosh u_0 \int \sqrt{h_0}\Bigg(h_0^{ij}(\partial_i y_3 \partial_j y_3 + \partial_i y_2 \partial_j y_2 + \partial_i y_m \partial_j \bar{y}_{\bar{m}})$$

$$- 2ky_3\partial_\xi y_2 + 3(3\cosh^2 u_0 - 2)y_3^2 - 2kr\tanh u_0\big((y_4 + \bar{y}_{\bar{4}})\partial_\xi y_2 + \partial_\xi(y_4 + \bar{y}_{\bar{4}})y_2\big)$$

$$+ \frac{ik}{4}\big(y_m \partial_\xi \bar{y}_{\bar{m}} - \bar{y}_{\bar{m}}\partial_\xi y_m\big) + \frac{ikr^2}{4}\big(y_4 \partial_\xi y_4 + \bar{y}_{\bar{4}}\partial_\xi \bar{y}_{\bar{4}}\big)\Bigg). \qquad (42)$$

---

[3]This is convenient in order to write the purely normal fluctuation coordinates in (40).



$r$ multiplies the last terms on both the second and third lines. Those are total $\xi$ derivatives and integrate to zero as the fluctuations are periodic. This means the dependence on $r$ drops out from the action.

The Wess-Zumino action is obtained by integrating the pulled back 3-form $C_3$ (8), obtaining at second order in the fluctuations

$$
S_{\text{WZ}}^{(2)} = \int {}^*C_3^{(2)} = \frac{R^3}{4} T_{\text{M2}} \cosh u_0 \int d^3\sigma \sqrt{h_0} \left( 3y_3^2 (3\cosh^2 u_0 - 2) - 3k y_3 \partial_\xi y_2 \right). \tag{43}
$$

Combining the two, we obtain the full quadratic bosonic action

$$
\begin{aligned}
S^{(2)} = \frac{R^3}{4} T_{\text{M2}} \cosh u_0 \int d^3\sigma \sqrt{h_0} \bigg( & h_0^{ij} \left( \partial_i y_m \partial_j \bar{y}_{\bar{m}} + \partial_i y_3 \partial_j y_3 + \partial_i y_2 \partial_j y_2 \right) \\
& + k y_3 \partial_\xi y_2 + \frac{ik}{4} \left( y_m \partial_\xi \bar{y}_{\bar{m}} - \bar{y}_{\bar{m}} \partial_\xi y_m \right) \bigg).
\end{aligned} \tag{44}
$$

Defining $\chi = y_3 + i y_2$, we obtain the same action for fluctuations in the 1/2 BPS case (21).

## 3.2 Fermionic action

To write the fermionic action in the 1/3 BPS case, we follow the logic as in Appendix A.2 of [23] and choose a local frame at the classical solution (38) and adapted to it. We use $e^{\hat{a}}$, $e^{\hat{3}}$, $e^{\hat{5}}$, $e^{\hat{\bar{5}}}$, $e^{\hat{6}}$, $e^{\hat{\bar{6}}}$ as in (22) and for the remaining four take

$$
e^{\hat{7}} = \frac{\sinh u_0 \tanh u_0}{2} d\phi + \frac{1}{2k(1+r^2)\cosh u_0} \left( ikr(e^{2i\xi/k} dw^4 - e^{-2i\xi/k} d\bar{w}^{\bar{4}}) - 2(r^2-1)d\zeta \right),
$$

$$
e^{\hat{2}} = -\frac{\tanh u_0}{2} d\phi + \frac{\tanh u_0}{2k(r^2+1)} \left( ikr(e^{2i\xi/k} dw^4 - e^{-2i\xi/k} d\bar{w}^{\bar{4}}) - 2(r^2-1)d\zeta \right), \tag{45}
$$

$$
e^{\hat{4}} = \frac{e^{2i\xi/k}}{1+r^2} dw^4 - \frac{2ir}{k(1+r^2)} d\zeta, \qquad e^{\hat{\bar{4}}} = \frac{e^{-2i\xi/k}}{1+r^2} d\bar{w}^{\bar{4}} + \frac{2ir}{k(1+r^2)} d\zeta.
$$

Note that $e^{\hat{7}}$ is tangential to the world-volume (as are $e^{\hat{0}}$ and $e^{\hat{1}}$) and $e^{\hat{2}}$, $e^{\hat{4}}$ and $e^{\hat{\bar{4}}}$ are normal.

The fermionic action requires the pullback of the vielbeine and with this choice we only need to check the norm of $e_{\hat{7}}$, which pulls back to

$$
{}^*e_{\hat{7}}^\xi = \frac{k}{\cosh u_0}. \tag{46}
$$

The world-volume gamma matrix $\gamma^\xi$, defined as in (27), then is

$$
\gamma^\xi = \frac{k}{2} \Gamma^{\hat{7}}, \tag{47}
$$

while the other two $\gamma^a$ are as in the 1/2 BPS case (28).

To complete the action we need to write the covariant derivatives (31), which requires the pullback of the spin connection. The definition in (45) is local and we cannot differentiate it to get the bulk spin connection. Instead, we get the pullbacks from the 11d Christoffel symbols for the metric (3) $\Lambda^A_{\ BC}$ using

$$
{}^*\Omega_i^{\hat{A}\hat{B}} = t_i^C e_A^{\hat{A}} \Lambda^A_{\ BC} e^{B\hat{B}} + e_A^{\hat{A}} \partial_i e^{B\hat{B}}, \tag{48}
$$

Here $t_i^C = \partial_i X^C$, with $X^C$ the target space coordinates (38), are tangent vectors and in particular

$$
t_\xi = \frac{2}{k} \partial_\phi + \frac{2ir}{k} \left( e^{-2i\xi/k} \partial_{w^4} - e^{2i\xi/k} \partial_{\bar{w}^{\bar{4}}} \right) + \partial_\zeta. \tag{49}
$$

For the $AdS_2$ directions, $^*\Omega_a{}^{\hat{A}\hat{B}}$ are the same as found from (24). The components in the $\xi$ direction are

$$^*\Omega_\xi^{\hat{m}\hat{\bar{n}}} = i\kappa^{\hat{m}\hat{\bar{n}}}, \qquad ^*\Omega_\xi^{\hat{2}\hat{3}} = \frac{2}{k}, \qquad ^*\Omega_\xi^{\hat{3}\hat{7}} = \frac{2\sinh u_0}{k}. \tag{50}$$

Plugging this into the action (26) and using the same kappa fixing condition as in (34)

$$\frac{1}{2}(1 + \Gamma_{\hat{1}\hat{2}\hat{7}})\theta = 0, \tag{51}$$

to replace $(1 - \gamma)\theta = 2\theta$, we immediately obtain

$$S_F^{(2)} = -\frac{R^2 T_{\text{M2}} \cosh^2 u_0}{2} \int d^3\sigma \sqrt{h_0}\, \bar{\theta} \left(\left(\gamma^a \partial_a + \frac{1}{2}{}^*\Omega_a^{\hat{0}\hat{1}}\gamma^a\Gamma_{\hat{0}\hat{1}}\right) + \frac{k}{2}\Gamma^{\hat{7}}\partial_\zeta \right. \\ \left. + \frac{5}{4}\Gamma^{\hat{0}\hat{1}\hat{2}\hat{3}} + \frac{1}{4}\Gamma_{\hat{3}}\sinh u_0 + \frac{i}{4}\kappa^{\hat{m}\hat{\bar{n}}}\Gamma^{\hat{7}}\Gamma_{\hat{m}\hat{\bar{n}}}\right)\theta, \tag{52}$$

which is the same as 1/2 BPS case (35), apart from the coefficient of the $\Gamma_{\hat{3}}$ term, but it anyhow vanishes by the kappa fixing condition (51).

Both the bosonic (44) and fermionic (52) actions agree with those of the 1/2 BPS configuration (21) (35) and in turn with those of [36] in the $k = 1$ and $k = 2$ cases. The results of the one-loop determinant are then the same in all those examples, as discussed in Section 1.

# Acknowledgements

We are grateful to S. Giombi, J. Maldacena and A. Tseytlin for helpful discussions. ND would like to thank CERN, EPFL and DESY for their hospitality in the course of this work.

**Funding information** ND's research is supported by the Science Technology & Facilities council under the grants and ST/P000258/1 and ST/X000753/1. OS's research is funded by the Engineering & Physical Sciences Research Council under grant number EP/W524025/1.

# A  Some properties of $\mathbb{CP}^3$

In this appendix, we discuss some of the properties of the space $\mathbb{CP}^3$ including its vielbein basis and spin connection. The space $\mathbb{CP}^3$ can be parameterised by three complex coordinates $(w^4, w^5, w^6)$ with the Fubini-Study metric as written in (6). For this metric, a choice of 3 complex vielbein, $e^{\hat{m}}$, with $\hat{m} = \hat{4}, \hat{5}, \hat{6}$, is [46]

$$e^{\hat{m}} = \frac{1}{\sqrt{1 + w^{\hat{m}}\bar{w}^{\hat{m}}}}\left[dw^{\hat{m}} - \left(1 - \frac{1}{\sqrt{1 + w^{\hat{m}}\bar{w}^{\hat{m}}}}\right)\frac{w^{\hat{m}}\bar{w}^{\hat{n}}dw^{\hat{n}}}{w^{\hat{m}}\bar{w}^{\hat{m}}}\right]. \tag{53}$$

The metric and Kähler form on unit radius $\mathbb{CP}^3$ is written in terms of the vielbeine as

$$ds^2_{\mathbb{CP}^3} = \kappa_{\hat{\bar{m}}\hat{m}}e^{\hat{\bar{m}}}e^{\hat{m}} + \kappa_{\hat{m}\hat{\bar{m}}}e^{\hat{m}}e^{\hat{\bar{m}}}, \qquad K = K_{\hat{m}\hat{\bar{n}}}e^{\hat{m}} \wedge e^{\hat{\bar{n}}}. \tag{54}$$

where $\kappa_{\hat{m}\hat{\bar{n}}} = \kappa_{\hat{\bar{n}}\hat{n}} = -iK_{\hat{m}\hat{\bar{n}}} = \delta_{mn}/2$.

The spin connection for $\mathbb{CP}^3$ in the basis $e^{\hat{m}}$ solves

$$de^{\hat{m}} + \hat{\Omega}^{\hat{m}}{}_{\hat{n}} \wedge e^{\hat{n}} = 0, \tag{55}$$

and is given by [46]

$$\hat{\Omega}^{\hat{m}}{}_{\hat{n}} = i(\omega^{\hat{m}}{}_{\hat{n}} - A\delta^{\hat{m}}{}_{\hat{n}}),\tag{56}$$

where $\omega^{\hat{m}}{}_{\hat{n}}$ is real and is given by

$$\begin{aligned}
\omega^{\hat{m}}{}_{\hat{n}} &= \frac{1}{i|w|^2}\left(1 - \frac{1}{\sqrt{1+|w|^2}}\right)(w^{\hat{m}}d\bar{w}^{\bar{\hat{n}}} - \bar{w}^{\bar{\hat{n}}}dw^{\hat{m}}) \\
&\quad + \frac{1}{2i|w|^4}\left(1 - \frac{1}{\sqrt{1+|w|^2}}\right)^2(\bar{w}^{\bar{\hat{k}}}dw^{\hat{k}} - w^{\hat{k}}d\bar{w}^{\bar{\hat{k}}})w^{\hat{m}}\bar{w}^{\bar{\hat{n}}},
\end{aligned}\tag{57}$$

and $A$ is defined in (7). We note that complex conjugation of (55) gives $\hat{\Omega}^{\hat{m}}{}_{\hat{n}} = -\hat{\Omega}^{\bar{\hat{n}}}{}_{\bar{\hat{m}}}$.

## A.1 $U(1)$ Fibre over $\mathbb{CP}^3$

The spin connection of the $U(1)$ fibre over $\mathbb{CP}^3$ solves

$$de^{\hat{7}} + \Omega^{\hat{7}}{}_{\hat{m}} \wedge e^{\hat{m}} + \Omega^{\hat{7}}{}_{\bar{\hat{m}}} \wedge e^{\bar{\hat{m}}} = 0.\tag{58}$$

Choosing $e^{\hat{7}} = (d\zeta/k + A)$ as we did in (22), one has $de^{\hat{7}} = dA = 2K = 2K_{\hat{m}\bar{\hat{m}}}e^{\hat{m}} \wedge e^{\bar{\hat{m}}}$ with the components $K_{\hat{m}\bar{\hat{n}}}$ as defined in (25). Therefore, we have

$$\Omega^{\hat{7}}{}_{\hat{m}} = K_{\hat{m}\bar{\hat{n}}}e^{\bar{\hat{n}}}, \qquad \Omega^{\hat{7}}{}_{\bar{\hat{m}}} = -K_{\hat{n}\bar{\hat{m}}}e^{\hat{n}}.\tag{59}$$

Finally, $\mathbb{CP}^3$ components of the spin connection of the total space can be similarly found to be

$$\Omega^{\hat{m}}{}_{\hat{n}} = -\Omega^{\bar{\hat{n}}}{}_{\bar{\hat{m}}} = \hat{\Omega}^{\hat{m}}{}_{\hat{n}} + i\delta^{\hat{m}}{}_{\hat{n}}e^{\hat{7}}.\tag{60}$$

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
