# Peer review of "Vortex loop operators and quantum M2-branes"

_SciPost Physics, doi:SciPost Phys. 17, 016 (2024)_

## Round 1 · Referee Report · Anonymous (Referee 1) · 2024-3-12

Strengths

1) interesting topic 2) concisely but clearly written 3) detailed computations, which can be easily be reproduced

Report

This paper tackles a general, important problem in holography, which is the computation of one-loop corrections of holographic duals of certain non-local operators. In this particular case, the gauge theory operators are 1/2 BPS and 1/3 BPS vortex loops in ABJM theory and their duals are M2 branes in AdS4xS7. The authors manage to show that the quadratic fluctuations around the classical M2 brane action reduce, for these operators, to previously known cases, whose results can then be recycled. This provides a new prediction for a (at the moment missing) localization computation, which would be worthwhile to attempt to try to match eq. (1.5), the main result of this paper.

---

## Round 1 · Referee Report · Anonymous (Referee 2) · 2024-5-25

Report

The authors study the one-loop corrections to the classical M2-brane action of configurations associated with BPS vortex loops opeators in AdS4 X S7/Z_k.
The paper is very short and direct to the point, the computation of fluctuations around such classical configurations. The result is then compared with the Wilson loop ones. There are some expectations for the form of the result but a QFT computation is still lacking. The extreme compactness of the paper is not necessarily a weakness and I found it refreshing and easy to read. An introduction, dense with information, describes the state of the art. This a sound and useful technical computation which deserves to be published.

Recommendation

Publish (meets expectations and criteria for this Journal)

---

## Editorial Decision

published